# Reduced Visual Magnocellular Event-Related Potentials in Developmental Dyslexia

**DOI:** 10.3390/brainsci11010048

**Published:** 2021-01-05

**Authors:** John Stein

**Affiliations:** Deptment Physiology, Anatomy & Genetics, University of Oxford, Oxford OX1 3PT, UK; john.stein@dpag.ox.ac.uk

**Keywords:** dyslexia, visual, magnocellular, parvocellular, timing, VERPs, spectral analysis, biomarker, hemifield, handedness

## Abstract

(1) Background—the magnocellular hypothesis proposes that impaired development of the visual timing systems in the brain that are mediated by magnocellular (M-) neurons is a major cause of dyslexia. Their function can now be assessed quite easily by analysing averaged visually evoked event-related potentials (VERPs) in the electroencephalogram (EEG). Such analysis might provide a useful, objective biomarker for diagnosing developmental dyslexia. (2) Methods—in adult dyslexics and normally reading controls, we recorded steady state VERPs, and their frequency content was computed using the fast Fourier transform. The visual stimulus was a black and white checker board whose checks reversed contrast every 100 ms. M- cells respond to this stimulus mainly at 10 Hz, whereas parvocells (P-) do so at 5 Hz. Left and right visual hemifields were stimulated separately in some subjects to see if there were latency differences between the M- inputs to the right vs. left hemispheres, and these were compared with the subjects’ handedness. (3) Results—Controls demonstrated a larger 10 Hz than 5 Hz fundamental peak in the spectra, whereas the dyslexics showed the reverse pattern. The ratio of subjects’ 10/5 Hz amplitudes predicted their reading ability. The latency of the 10 Hz peak was shorter during left than during right hemifield stimulation, and shorter in controls than in dyslexics. The latter correlated weakly with their handedness. (4) Conclusion—Steady state visual ERPs may conveniently be used to identify developmental dyslexia. However, due to the limited numbers of subjects in each sub-study, these results need confirmation.

## 1. Introduction

The most widely accepted view of the cause of developmental dyslexia is that it results from failure to acquire the ability to translate visual symbols, letters, into the sounds, phonemes, they represent [1] However, this phonological theory does not explain why this skill does not develop in some children. The magnocellular hypothesis of dyslexia was introduced by Lovegrove [2] to fill this gap. He had found that dyslexics had reduced sensitivity to low-spatial-frequency and high-temporal-frequency visual grating patterns, suggesting a deficit in the visual “transient system”, which is responsible for timing visual events [3]. This failure of timing could explain failure to see words properly, hence failure to distinguish their component sounds properly in the right order as well. Lovegrove’s hypothesis has since been confirmed many times [4]. Livingstone et al. [5] showed how his psychophysical results might be explained by impaired development of the visual timing cells in the brain, the visual magnocellular neurons. They found in five brains of dyslexic people examined post mortem, the magnocellular (M-) cells in the visual relay in the thalamus, the lateral geniculate nucleus (LGN), were considerably smaller than normal whereas the parvocells were unaffected. Recently, these observations have been confirmed in vivo by seven-tesla magnetic resonance imaging (MRI) experiments, whose improved resolution now allows measurement of the size and shape of the LGN in living people [6].

Since that time, the evidence that many dyslexics have mildly impaired development of this visual magnocellular/transient processing system has become much stronger, although it still remains controversial [7]. Psychophysically, many dyslexics are found to display lower flicker fusion frequencies [8], reduced motion sensitivity [9], impaired focussing of visual attention [10], abnormal metacontrast masking [11] and unsteady binocular control [12]. Thus, the development of visual M- cells seems to be impaired in many, but probably not all, dyslexic people.

Livingstone et al. [5] also recorded transient visual event-related brain potentials (VERPs) elicited by flickering stimuli in some dyslexic subjects. Such low-spatial-frequency and high-temporal-frequency gratings selectively stimulate the magnocellular system, and they found that their dyslexics’ VERPs were reduced and delayed compared with controls’. Although many studies have confirmed their findings [13], some have not. For example, Victor et al. [14] studied only ten dyslexics selected merely on the basis of their reading backwardness. Two of these had suffered reading problems as a result of a head injury, so they were unlikely to exhibit a neurodevelopmental magnocellular deficit anyway.

A simple VERP-based test for developmental dyslexia could transform the current confusion surrounding its diagnosis by providing an objective biomarker not dependent upon controversial psychometrics [7]. We therefore set out to test whether reduced magnocellular function can be detected reliably from steady state VERP recordings.

## 2. Methods

### 2.1. Acronyms

BAS—British Abilities ScalesEEG—electroencephalogramLGN—lateral geniculate nucleusMRI—magnetic resonance imaging PegQ—pegboard quotientVERP—visual event-related potential

### 2.2. Subjects

This report summarises recordings from several different studies made over a number of years in a total of 48 male and 20 female dyslexics aged between 16 and 25, compared with 88 (31% female) controls, matched also for age and socioeconomic status. Although the individual studies were small, amalgamating them for this paper has enabled their power to increase.

Ethics: The protocols for these studies were all reviewed, and permission to carry them out was granted by the Oxford Psychiatric Research Ethics Committee (OPREC O 01.02). All participants read and discussed the protocols and signed their consent to participate in them.

The dyslexics had all been diagnosed as such by qualified psychologists on the basis of a difference of at least 1.5 standard deviations (sds) between their nonverbal ability (some version of a matrices test) and their single-word reading scores, and summaries of these reports were reviewed by us. This “discrepancy” definition of dyslexia is now controversial, but de facto most practitioners still use the difference between subjects’ reading and nonreading skills as their main criterion for the diagnosis. Many of the subjects were also tested by us on the British Abilities Scales (BAS), from which we calculated the discrepancy between their matrices and single-word reading (Reading z—Matrices z scores). Additionally, in some of our dyslexics and controls we administered the Castles and Coltheart regular, irregular and nonword reading tests [15].

In order to measure the subjects’ hand skill, many also carried out the Annett pegboard task [16]. For this, we recorded the time taken to move a row of pegs from one set of holes to another with the left (L) and right (R) hands separately. From these times we derived a measure of relative hand skill, peg quotient (PegQ) = (L − R)/((L + R)/2), which adjusts for overall differences in hand skill between subjects. This provides an approximately normally distributed variable with a positive mean. A positive PegQ indicates superior right-hand skill relative to the left, and a negative PegQ suggests superior left-hand skill.

### 2.3. Stimuli

The subjects were asked to look at a spot situated at the centre of a visual display unit (VDU) placed 57 cm in front of them, without moving their eyes. The screen displayed a large (20 × 20°) chequer board whose checks switched from black to white and vice versa (contrast reversals) over a period of 200 ms. Hence, the fundamental frequency of the stimulus was 5 Hz, and the 2nd harmonic was at 10 Hz. These stimuli made the squares appear to flicker and to move from side to side. Each check was 2 × 2 cm in size (2° at the viewing distance of 57 cm) so that the predominant spatial frequency in the stimulus was 0.5 cycle per degree. Although lower than optimum to stimulate magnocellular neurons, this low frequency is the standard clinically and definitely stimulates M- cells more than parvocells (P-) [17]. We used checks whose spatial contrast changed sinusoidally in some experiments to eliminate the higher spatial frequencies generated by the check edges, but this made no difference to the amplitude or phase of the fundamental and 2nd-harmonic responses reported here. The contrast (white − black)/(white + black) of the checks was either 50% (high) or, for many of the subjects, 10% (low).

In order to investigate the visual activation of the left and right hemispheres separately to compare with their handedness, many of the subjects were also asked to fixate on spots positioned 2.5 cm (i.e., 2.5°) to the left or right of the screen. This procedure separately stimulated the right or left parafoveal hemifields, avoiding the central foveal 5° of the visual field. Since there are far fewer magno- than parvocells in the fovea [18], avoiding stimulating the fovea prevented parvocell responses from dominating these recordings.

### 2.4. Recording

Clinically, the standard way of recording VERPs is to use a contrast-reversing chequer board stimulus as here [19]. However, the dominant P100 recorded from the centre of the back of the occiput (O zentrum—Oz—in the International EEG 10–20 system) is an amalgam of responses set up in the right and left hemispheres by stimulation of the left and right hemifields, respectively. The earliest components of these are mirror images of each other and therefore tend to cancel each other out at scalp location Oz [20]. Therefore, in order to maximise the responses, we have recorded the interoccipital VERP between left (O1) and right (O2) occipital leads.

In addition, rather than recording transient VERPs, we recorded the subjects’ steady state VERPs (SSVERPs) because they are easier to record, highly reproducible and their power and phase at different frequencies can be measured very precisely.

Unfortunately, we had to use 4 different recording systems over the different studies. Although we always recorded between O1 and O2, i.e., with approximately the same electrode placements, we used 4 different amplifiers and sets of electrodes with different properties for different projects, which meant that the noise and amplitude of the recordings varied greatly because of the different setups.

### 2.5. Analysis

The interoccipital EEG was recorded between left and right occipital poles (O1 and O2) with the left mastoid grounded. If the subject’s eyes moved during a recording this caused a large artefact and that epoque was discarded. A total of 256 epoques of 1 s, covering 5 full cycles, were averaged (pass band 1–300 Hz, sampling rate 1000 per second). Then, the fast Fourier transform was used to compute the amplitude and phase lag of each frequency in the averaged 1 s signal.

M- cell responses are nonlinear in the sense that they respond equally to dark/light and light/dark transients [21]. Thus, they set up steady state-evoked potential responses with larger peaks at the 2nd harmonic (10 Hz for our stimulus) than at the fundamental (5 Hz). However, the linear parvocell (P-) system responds more at the fundamental frequency. At low contrast, both were activated less. Since the black/white and white/black reversals occurred at the 2nd harmonic, the predominant response of the M- system was expected to be at 10 Hz and at 5 Hz for the P- system. There would have been a small contribution of the M-system at 5 Hz, but the main contributor at the fundamental was likely to be the P- system.

We therefore focused on measuring the fundamental (5 Hz) and 2nd-harmonic (10 Hz) components’ amplitude and phase in the steady state spectra to distinguish the contributions of the P- and M- cells, and we calculated the ratio of the second harmonic to the fundamental, because this served to normalise individual differences in EEG amplitudes due to variations in electrode positioning and the resistances of electrodes, scalp and skull, and so forth.

## 3. Results

These dyslexics still had very significant reading problems. Whilst 36 controls made an average of 1.6 (sd 2.4) errors in the Castles and Coltheart irregular word reading test and 2.8 (2.6) errors in the nonword test, 27 dyslexics made many more errors (8.2 (5.1) on irregular words and 8.8 (5.1) on nonwords). These differences were statistically significant (*p* < 0.01 for all the reading tests). There was also a trend (ns) for the dyslexics to be less right-handed on the PegQ test than the controls.

Figure 1 shows an amplitude spectrum of the steady state interoccipital VERP from a control subject in response to full-field chequer board stimulation. The largest peak is at the second harmonic check reversal frequency of 10 Hz, but peaks can also be seen at the fundamental (5 Hz) and at the 4th and 6th harmonics. The higher harmonic powers decreased greatly when we used sinusoidally contrasting checks, but this made very little difference to the fundamental and second-harmonic amplitudes.

Figure 2 shows the very different spectrum of a dyslexic. This is dominated by a peak at 5 Hz with a smaller peak at 10 Hz. Thus, the 10 Hz peak was larger in the control, but the 5 Hz peak was larger in the dyslexic subject.

Figure 3 shows the average amplitude of the 5 and 10 Hz components in the dyslexics and controls in response to the high-contrast stimulus. These, particularly the dyslexics’, have very large SDs due to inter-individual variability, caused by different electrode resistances, recording systems, montages, scalp and skull thicknesses varying greatly. Hence, neither of the differences between control and dyslexics responses were statistically significant at either 5 or 10 Hz.

In order to allow for the different gains and resistances discussed above, we normalised the amplitudes by calculating the ratio of 10/5 Hz amplitudes in each subject for both high- and low-contrast stimuli. Figure 4 shows that this ratio averaged only 0.57 in the dyslexics compared with 1.67 in the controls, i.e., it was nearly three times higher in the controls. This difference was highly statistically significant (*t* (154) = 3.54, *p* < 0.001). Note that it was driven by both reductions in the 10 Hz amplitude and also increases in the 5 Hz amplitude in the dyslexics.

### 3.1. Correlation with Reading

Since we had postulated that impaired visual function may be a contributory cause of impaired reading, we wished to see if individuals’ 10/5 Hz ratio predicted their reading scores. For 52 poor readers (both dyslexic and less-severe poor readers) for whom we had all the data, we correlated their 10/5 Hz ratio with the discrepancy between their single-word reading and their intelligence z scores. The results are shown in Figure 5. The higher the ratio, the smaller the discrepancy (*r* = 0.59, *p* < 0.01).

### 3.2. Latency

Since magnocellular ganglion cells are smaller in dyslexics, their axonal conduction velocity should be reduced compared with controls. Therefore, we expected the second harmonic (10 Hz) component to reach the primary visual cortex more slowly in our dyslexics. We therefore calculated the average latency of the second-harmonic peak from its phase with respect to the chequer board reversals; this averaged 33 ms in the controls and 37 ms in the dyslexics, i.e., that of the dyslexics was 4 ms longer. However, this difference did not reach statistical significance (*p* < 0.15).

### 3.3. Hemifield Stimulation

In many of the subjects, we recorded VERPs not only in response to whole-field stimulation when the subjects were fixating on the centre of the screen, but also during hemifield stimulation avoiding the central 5° (foveal) field. Stimulating the right hemifield activated the left hemisphere first and vice versa. Following left hemifield stimulation (passing first to the right hemisphere), the amplitude of the 10 Hz interoccipital response averaged 20.62 microvolt/Hz. This was significantly larger than that following right hemifield stimulation (18.13 microvolt/Hz—F(1,66) = 4.02, *p* < 0.05). The delay between each check reversal and the second-harmonic responses in the primary visual cortex was calculated from the phase data. That in the right hemisphere following left half-field stimulation averaged 27 ms, whereas that in the left visual cortex following right half field stimulation was 38 ms, a highly significant difference between the hemifield conditions (F(1,56) = 12.4, *p* < 0.001). Thus, the right hemisphere seems to enjoy an advantage in both amplitude and latency over the left hemisphere for its 10 Hz magnocellular input. However, for the 5 Hz component, there were no significant differences between either hemisphere or group.

To examine whether the dyslexics’ degree of lateralisation was reduced as has been suggested, we also calculated the interaction between group and the latency difference between left and right hemifield stimulation (condition). Although there was a trend for the dyslexics to show a smaller difference between left and right hemisphere latencies, the interaction between group and hemifield condition was not significant.

### 3.4. Handedness

In order to see how hemisphere differences might relate to the subjects’ handedness, we also correlated individuals’ hemifield VERPs with their degree of right-handedness. For the dyslexics, the more right-handed they were in the Annett pegboard test (PegQ), the larger was the 10 Hz second-harmonic component in their VERPs (*n* = 47, *r* = 0.18, *p* = 0.048). But the relationship between the two was only a trend in 34 controls.

## 4. Discussion

There seem to be no other reports of steady state VERP interoccipital recordings in dyslexics for comparison with these results. However, the present findings suggest that many dyslexics have a reduced second harmonic response to a reversing checker board stimulus, compared with control good readers, but demonstrate a larger response at the fundamental frequency. This pattern leads to dyslexics having a reduced second harmonic/fundamental ratio. It confirms our previous results [22] and is consistent with many results from other labs using transient VERP recordings, as summarised in Schulte-Korne et al. [13]. They support our hypothesis that many developmental dyslexics have impaired development of the visual magnocellular system, since input from this system seems to dominate the second harmonic component in the VERP. The relatively weak 10 Hz response in dyslexics probably underlies their problems with visual timing and sequencing that cause many of their difficulties with learning to read [4]. In further support of this view, it has also been found that this frequency plays a particularly important part in recruiting the attentional and eye movement control networks for reading [23].

### 4.1. Parvocellular 5 Hz Response in Dyslexia

Our participants with dyslexia seemed to have larger 5 Hz fundamental frequency responses than the controls, although this difference was not statistically significant, probably because the different set ups we used introduced increased “noise” into our comparisons. However, there are many reports that suggest that dyslexics may develop a greater density of parvocells than good readers. Parvocells are concentrated in the fovea and are responsible for signalling high spatial frequencies (fine detail) and colour, rather than timing. Hence, Lovegrove [2] found that under static conditions, dyslexics had a higher contrast sensitivity than controls at high spatial frequencies (>10 cycles per degree), and many researchers have shown that they have better acuity in the peripheral retina [24,25]. Likewise, Dain et al. [26] found that poor readers had significantly higher blue/yellow colour sensitivity than good readers. Thus, there is much evidence that dyslexics tend to have superior parvocellular function. During development, 75% of retinal ganglion cells that are born undergo programmed cell death because they fail to make useful connections [27]. If, under genetic control, neurogenesis of magnocells is reduced in dyslexics, then they will probably grow more parvocells and a higher proportion of these than usual will be able to make successful connections and survive. Thus, the reduced magnocellular numbers will populate the retina more sparsely, and this may allow increased numbers of parvocellular neurons to develop and fill the gaps, particularly in the periphery.

### 4.2. Right Hemisphere Advantage

Our hemifield stimulation experiments demonstrated a clear right hemisphere advantage in both latency and amplitude for the 10 Hz component in the VERP, but not for the 5 Hz component, and there was a trend for this to be reduced in dyslexics. Such a right hemisphere advantage has been shown for motion stimuli by several authors, for example, Niedeggen [28], but ours is the first report to link this specifically with the magnocellular (10 Hz) input. Nevertheless, Bosworth [29] showed greater effects of spatial attention in the left visual field input to the right hemisphere, specifically for motion processing by the visual striatofugal dorsal stream whose main input is supplied by the M- system. Thus, our and their results support the suggestion that initially the right hemisphere receives a larger magnocellular input that results in superior motion processing in the left visual hemifield. The trend we saw allows us to speculate that this advantage may be reduced in dyslexics as a consequence of their reduced magnocellular sensitivity.

### 4.3. Handedness

The more right-handed the dyslexic subjects were according to their PegQ, the larger was the 10 Hz (magnocellular) component in their VERPs. This was not found in the controls, probably because the handedness of good readers, whether to the left or right, tends to be stronger than in poor readers. This is because learning to read forces usually the left hemisphere towards greater commitment to left hemisphere dominance for communication, hence stronger right-handedness [30,31]. Since the dyslexics showed a trend to be less right-handed than the controls, i.e., more ambidextrous, this fits with other literature that suggests that the less ambidextrous you are the better you are likely to be at reading [16].

### 4.4. Correlation with Reading

We found correlations between 52 of our subjects’ magnocellular sensitivity as indexed by the ratio of their 10/5 Hz responses, and how far their reading abilities lagged behind their nonverbal abilities as measured by the BAS matrices test. This confirms several other studies that have demonstrated an association between magnocellular function and reading, measured in many different ways [4]. Nevertheless, a phonological deficit is still widely considered to be the most important cause of dyslexic reading difficulties. Yet the results reported here suggest that the majority of these dyslexics demonstrate reduced visual magnocellular coupled with increased parvocellular sensitivity as well, and that the size of this difference predicts their reading deficits.

### 4.5. Limitations

The greatest limitation of these studies is that not all the procedures were carried out on all the subjects. It is an amalgam of several small studies that were individually underpowered. They have been grouped together to try to draw out some common themes. Inevitably this means that there is more “noise” in this data. Nevertheless, one could argue that, given this weakness, the robust finding that the ratio of the second harmonic, magnocellular, 10 Hz component in the SSVERP divided by the 5 Hz component is very significantly smaller in dyslexics is even more impressive.

Another limitation was that the chequer board stimulus is not optimal for selectively activating the M- system. However, we used it for a number of good reasons. First, it is the standard for recording visual ERPs clinically; it elicits large responses that are well understood and there is much experience of using it. Subjectively, this stimulus appears both to flicker and to move and it elicits strong M- stream responses. Finally, the phase-reversing chequer board elicits clear and easily computed fundamental and second harmonic components in the averaged steady state potential, and these were the responses we wanted to quantify.

## 5. Conclusions

These results suggest that visual magnocellular responses may be reduced in dyslexics whilst the parvocellular ones are increased. This imbalance may be a contributory cause of dyslexics’ reading problems. As M- impairment can lead to the inefficient visual temporal processing, poor focussing of visual attention and perceptual instability that is seen in many dyslexics, measuring people’s 10/5 Hz ratio could provide us with a physiological biomarker of developmental dyslexia. Reliable EEG recording equipment is now available at reasonably low prices to do this. Hence, recording the SSVERP 10/5 Hz ratio routinely in people with reading difficulties could enable us to diagnose dyslexia cheaply, conveniently and quickly. But the findings reported here need confirmation in a larger study in which all the tests described here are carried out in all the subjects.

## Figures and Tables

**Figure 1 brainsci-11-00048-f001:**
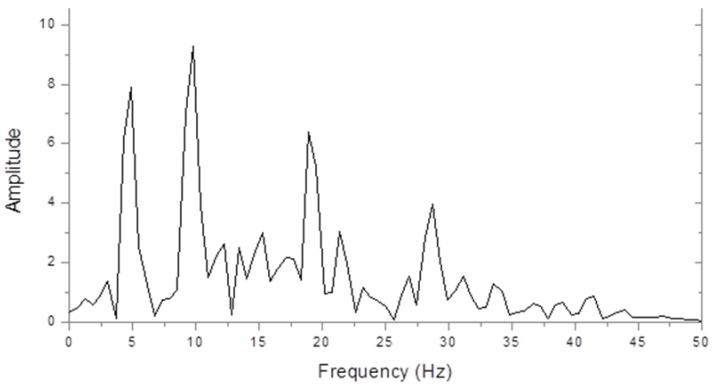
Control steady state visually evoked event-related potentials (VERP) spectrum (microvolts/hertz) in response to chequer board stimulation at 5 Hz.

**Figure 2 brainsci-11-00048-f002:**
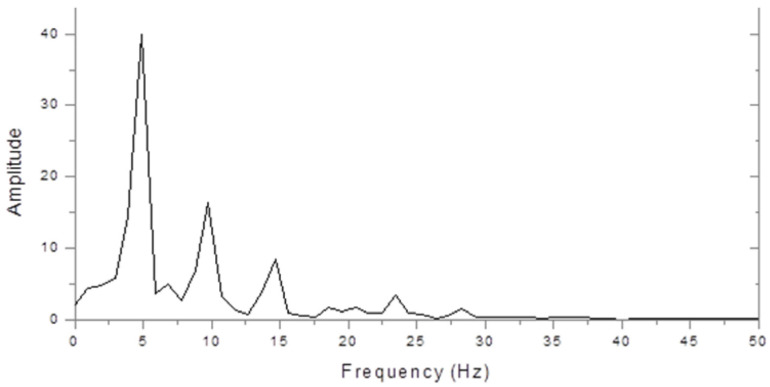
Dyslexic steady state VERP spectrum (microvolts/hertz).

**Figure 3 brainsci-11-00048-f003:**
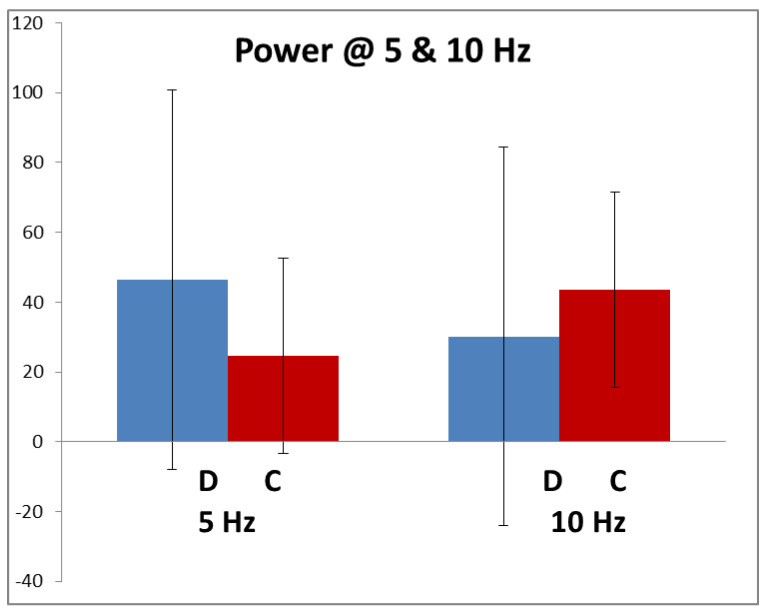
Average amplitude (microvolt/hertz) at 5 Hz and 10 Hz (+/−1 sd). Dyslexic *n* = 68; control *n* = 88.

**Figure 4 brainsci-11-00048-f004:**
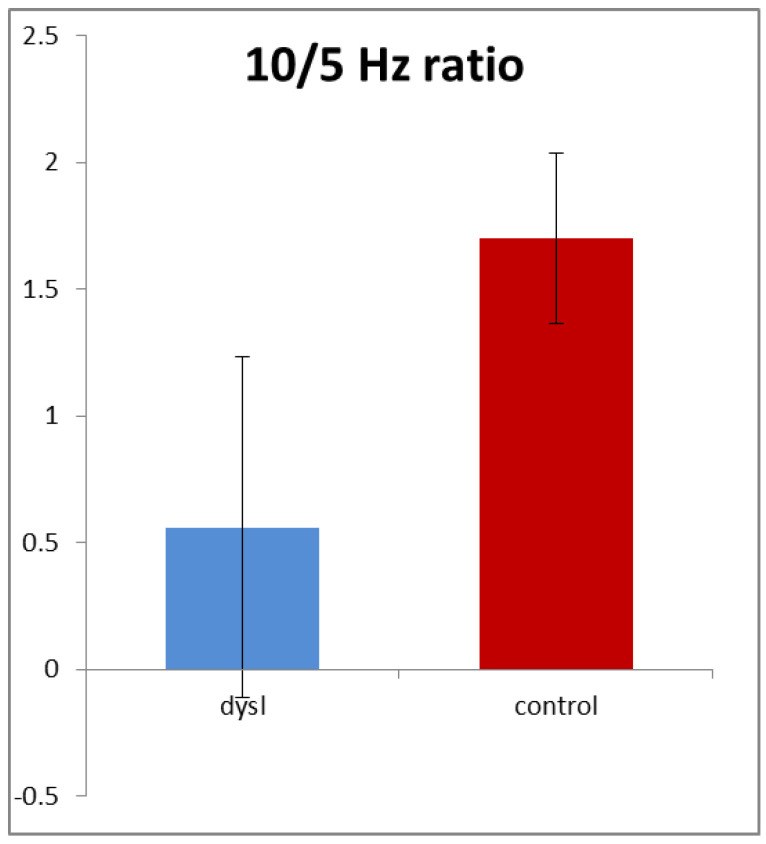
The 10/5 Hz ratio (+/−1 sd). Dyslexic *n* = 68; control *n* = 88.

**Figure 5 brainsci-11-00048-f005:**
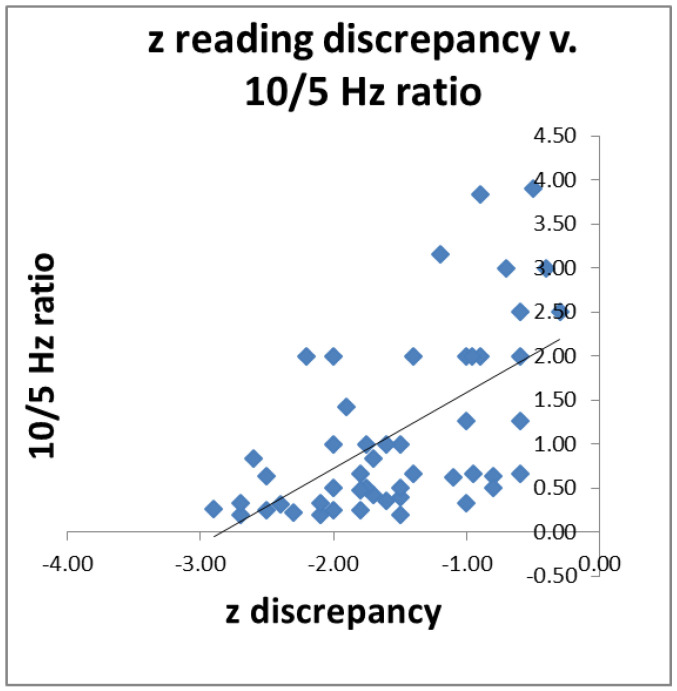
Reading discrepancy was predicted by 10/5 Hz ratio in 52 poor readers; *r* = 0.59, *p* < 0.01.

## Data Availability

The data presented in this study are available on request from the corresponding author. The data are not publicly available due to patient confidentiality.

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
