# Peer review of "Reduced Visual Magnocellular Event-Related Potentials in Developmental Dyslexia"

_brainsci, 2021, doi:10.3390/brainsci11010048_

Round 1

Reviewer 1 Report

In this article, the author describes the results of multiple small researches on magnocellular deficit that could characterize adults with a reading disorder. As described in the limitation section, the greatest limitation of the article is that results come from multiple studies, and in these, the variables collected and analyzed were not always the same. Anyhow, the difference observed between adults with and without reading disorder in the 10/5 Hz ratio observed appears an interesting result, that could be a starting point for future confirming research.

Detailed comments:

  • Please introduce all the acronyms (e.g. VDU, LVF)
  • Page 1 Line 26 "His hypothesis has since been confirmed many, many times." please add literature
  • Considering that data about reading abilities in table 1 do not describe the entire sample, I think that the information in table 1 could be described inside the text, whereas information about the entire sample mean age, gender and SES should be described inside the table.
  • page 3 lines 115-118 "...M-cells responses are non-linear... ... responds more at the fundamental frequency" please add literature.
  • Quality of Figure 1 ad 2 is poor. Considering that these figures are only examples, the quality should be improved, or the figures could be removed.
  • About Figure 5, could be useful to distinguish between participants with and without reading difficulties.
  • I'm not able to find information about the number of participants used to calculate the correlation between Handedness and the 10 Hz harmonic component. I think that could be useful to define it in the Participants or Results section.
  • I think that in the Limitation (or Conclusion) section should be added a sentence to describe that additional researches that use these paradigms should be conducted to confirm the observed difference in the 10/5 Hz ratio.

Author Response

Ref 1

Thank you for your very useful suggestions:

Acronyms are now listed at the beginning of methods and also when each is introduced.

l. 26 Ref added.

Table 1 has been removed and the data introduced in the text as suggested.

P.3 - M cell responses’ non linearity Lennie ref. added.

I believe figures 1 & 2 are now much clearer.  It is necessary to show these for the reader to be able to see the clear difference in the dyslexic spectrum compared with the control.

Fig 5.  Numbers clarified.

Handeness v. 10 hz correlation numbers now reported.

I agree strongly about the need for these results to be confirmed.  Thank you.  This is now emphasised in the Abstract, Discussion and the Conclusions.

Reviewer 2 Report

This is an interesting and well written paper that uses visually evoked potentials to investigate impairments to visual timing systems in the brain as a biomarker of dyslexia. The results show a relationship between the peak amplitude and latency of key components and reading difficult, and this is taken to support accounts in which at least some people with dyslexia have impairment to visual processing associated with deficits in the magnocellular pathway. 

Claims for M-pathway deficits in dyslexia remain controversial but important. The present study provides interesting new evidence from a large sample of participants that is consistent with this claim. The manuscript is clearly written and the study appears to have been conducted well. Below I ask that some elements of the method are expanded upon for the benefit of readers who are unfamiliar with these methods. I would also ask that a sentence and some references is added to the introduction to note that the dominant view is that dyslexia is caused by deficits in phonological rather than visual processing. It is, however, fair of the author to argue that there is also evidence for visual deficits. 

Minor points

line 57. Everyone wants to know about statistical power these days. It would be helpful to include a comment explaining why this sample size was selected. It's larger than I had anticipated and seems plausible, but worth commenting on.

line 81 - something's gone wrong with the degree symbol

lines 86 & 95 - what's the evidence that 0.5 cycles of spatial frequencies preferentially stimulates magnocells?

lines 92-97. While I don't dispute that this method would selectively project stimuli to parafoveal hemifields, it would be helpful to report the eccentricity of the presentation in visual degrees (2.5cm = approximately 2.5 degrees at a 57cm viewing distance?). It would be helpful to include a sentence and reference(s) to explain parafoveal versus foveal areas and the distribution of parvocells versus magnocells for readers.

line 99. reference?

line 104-5. Is this routinely in previous research? if so, please reference.

line 119. report M-cells or magnocells consistently 

lines 119-126. This detail about the function of magnocells and parvocells is interesting but readers would benefit from direction to some reading about it using a reference. 

Author Response

Ref 2.  Thank you for your very helpful remarks, which have improved the paper considerably.

Introduction now explains how the phonological and magnocellular theories fit together.

Line 57 – statistical power emphasised.

  1. 81 – corrected
  2. 92 – New Reference and clarification of why the central 5 degrees of the visual field was not stimulated.
  3. 99 Reference added.
  4. 104 – Quite right. Interoccipital recording is not routine!
  5. 119 – Only magnocellular or M- cells now used throughout the paper
  6. 119 – Ref to non linearity of M- responses now included.